# Price Elasticity of Production Factors in Beijing's Picking Gardens

**Na Du**, **Qianqian Shao** **and Ruifa Hu** *

School of Management and Economics, Beijing Institute of Technology, Beijing 100081, China;
gudulenggongzhu@163.com (N.D.); shaoqianqian@gmail.com (Q.S.)
* Correspondence: ruifa@bit.edu.cn

**Abstract:** Picking agriculture is a form of leisure agriculture based on the concept of traditional garden. Due to their unique layout and construction style, picking gardens have different attractive elements, including sightseeing, leisure, entertainment, crop production, and crop picking. However, despite its increasing importance, there is no systematic research on price elasticity or price substitution elasticity of production factors in picking gardens. To fill this gap, we surveyed 308 farmers in five districts of Beijing and employed a translog cost function to compare the impact of operation patterns on peach and cherry production cost by estimating elasticities of substitution between and among inputs. We found that own-price elasticity of all input factors was negative, while substitution relationships existed between labor and land, labor and fertilizer, fertilizer and manure, and manure and pesticide. This indicates that Beijing's agricultural sector is labor intensive, while fertilizer and pesticide are scarcely used.

**Keywords:** picking agriculture; factors of production; translog cost function; price elasticity

---

## 1. Introduction

With an increase in income, leisure time, and consumption levels, urban residents' demands regarding their environments and leisure travel have increased. Indeed, it has become fashionable to return to greenery and nature [1,2]. Furthermore, with the integration of rural and urban environments, and agricultural and non–agricultural industries, urban agriculture has emerged as the need of the hour. A significant difference between urban agriculture and traditional agriculture is the relevance ascribed to leisure agriculture. The basis of urban agriculture is meeting the needs of citizens' consumption and urban development, and the goal is to increase farmers' income [3].

Leisure agriculture is a new type of agriculture production and management system that combines agriculture and tourism. It uses pastoral landscapes, agricultural production, and management activities, as well as the natural environment in rural areas, to attract tourists who wish to simply observe and relax. Leisure agriculture combines gardening, tourism, and garden production and harvesting, thus providing a combination of economic, ecological, and social benefits [4]. The economic benefits are reflected mainly in the promotion of rural industries, structural optimization, and upgrades. The development of leisure agriculture also brings direct economic benefits to its operators and enriches agricultural value. It allows tourists to engage in sightseeing and picking tours and to enjoy the harmonious unity of human and nature. Being in the countryside ensures tourists' social benefits, the desire to return to nature, the experiences in agriculture, and the improvement of their health [5].

Picking agriculture is one of the forms of leisure agriculture [6–8]. Studying the development of leisure agriculture tourism and understanding its comprehensive social, economic, cultural, and environmental benefits is important in promoting the development of the national economy

and improving in people's living standards and lifestyles [8,9]. Sustainable development of leisure agriculture can also guarantee higher incomes for farmers [7,10].

As leisure agriculture develops rapidly in China, its importance has been emphasized by several scholars. Among the current studies on the topic, most of them are descriptive in the form of discussions on its concepts, functional characteristics, new development patterns, and more. Qi and Zhu show that the development of leisure agriculture plays a comprehensive part in furthering the interests of society, economy, culture, and the environment at large; it also plays a very important role in promoting the development of the national economy, improving people's living standards, and changing lifestyles for the better [9]. Cheng and Cai and Zheng list several objectives of leisure agriculture, such as providing recreation, narrowing the rural–urban disparity, improving agricultural productivity, and offering healthcare [11,12]. Comparing with the studies on leisure agriculture, researches focused on the agricultural productivity and the input–output elasticities were abundant. Since Solow (1957) constructed the total production function to explore economic growth, the related research on establishing the production function to analyze economic growth determinants from the perspective of input–output have been numerous [13]. Vincent measured the elasticity of substitution between land, labor, and capital using Australia's 1920–1969 overall data, showing that there was a higher substitution between labor and capital and a clear complement between land and capital [14]. Griliches studied the input and output elasticity of production factors in the United States by using the Cobb–Douglas agricultural production function, and showed that the sum of the output elasticity of US production factors is around 1.2 with increasing returns to scale [15]. Yuize applied panel data to measure the output elasticity in developing countries such as China, and showed that the output elasticity of labor is higher than that of working animals, machinery, and fertilizers [16]. Hayami determined that the output elasticity of the Japanese agricultural labor input is around 0.5, while the output elasticity of the land is 0.2 smaller than that of the labor [17]. Based on the above research, Echevarria estimated the output elasticity of agricultural inputs in 127 countries and found that the output elasticity was only about 0.2 [18]. Haley and Lio established the agricultural total factor production functions to estimate the effect of inputs on outputs, and showed that the main factor determining agricultural growth is labor input, which is much higher than the contribution of land [19,20].

Chinese research on the contribution of agricultural input factors to agricultural growth started after the reform and opening up. From 1978 to 1984, the increment of agricultural productivity benefited from the household responsibility system and land contact system [21]. Provincial data showed labor and capital inputs contributed largely to agricultural growth [22]. Xin and Qin have argued that the output elasticity of capital input is higher than that of labor and land. Capital made a critical contribution to agricultural growth [23]. Moreover, Zhang found that agricultural intermediate investment and the fixed asset investment contributed the most to agricultural growth [24]. But China's agricultural growth has changed from capital-oriented input to technical advancement, according to Tao and Liu [25]. Besides studies on agricultural growth, factor allocation was also emphasized in the literature. The studies on input (e.g., fertilizer, machinery, plastic film, etc.) productivity from the perspective of factor allocation show that optimized input ratios are the key factors determining agricultural productivity, which can be conducted through education, social security systems, and the improvement of land contracting rights [22,26].

Even though the previous literature studied some of the key agricultural inputs, few works have covered systematic studies on the output elasticity of agricultural input factors and substitution elasticities, especially the substitution of fertilizer for manure. In this paper, we conduct a transcendental logistic (translog, henceforth) cost function, and apply seemingly unrelated regression (SUR) to analyze the input costs and cost shares of leisure agriculture in Beijing. Our study enriches leisure agriculture literature and offers policy implications for improving the input allocation efficiency of leisure agriculture and improving the income of urban farmers in Beijing.

## 2. Data and Methods

### 2.1. Functional Form

Most studies employing translog cost functions use the "own" elasticity of production factors to measure the magnitude of the rebound effect. This paper assumes the production function to be second-order differentiable, and uses Shephard's lemma to obtain cost–share functions, before estimating the own-price and cross-price elasticity of each input. According to demand–supply theory, efficiency changes in goods and services consumption are closely related to price functions. At the micro level, an increase in production factor prices often causes more capital investment in energy production. By contrast, a reduction in factor prices means households can achieve more profits at a lower input cost. The main inputs of agricultural production include capital ($K$), labor ($L$), and other input factors ($M$). Thus, the production function can be written as

$$Q = Q(K, L, M) \tag{1}$$

There is a duality relationship between production function and cost function, and, according to the duality theory, input-factor prices are exogenous. The cost production is given below.

$$C = C(P_k, P_l, P_m, Q) \tag{2}$$

where $Q$ represents the total output, $C$ the total cost, and $P_k$, $P_l$, and $P_m$, represent the prices of capital, labor, and other input factors, respectively. We construct the minimum-cost function corresponding to Equation (3). To reflect the relationship between factor price and input factor under the condition of cost minimization, this paper assumes the cost function to be a translog cost function. Its general form is

$$\ln C = \beta_0 + \sum_{i=1}^{m} \beta_i \ln P_i + 0.5 \sum_{i=1}^{m} \sum_{j=1}^{m} \beta_{ij} \ln P_i \ln P_j + \beta_y \ln Y + 0.5 \beta_{yy} (\ln Y)^2 + \sum_{i}^{m} \beta_{iy} \ln P_i \ln Y \tag{3}$$

where $C$ is cost, $Y$ is output, $P_i$ is the price of factor $i$, $P_j$ is the price of factor $j$, and $\beta_i$ is the coefficient to be estimated. The symmetry condition implies that $\beta_{ij} = \beta_{ji}$. Another restriction on the parameter estimates is that the cost function must be homogeneous of degree 1 in input prices given $Y$. This implies the following restrictions on Equation (4):

$$\sum_{i=1}^{n} \beta_i = 1, \ \sum_{i=1}^{n} \beta_{ij} = \sum_{j=1}^{n} \beta_{ji} = \sum_{i=1}^{n} \beta_{iy} = 0 \tag{4}$$

The translog cost function can be estimated directly, and gains in efficiency can be obtained by estimating the optimal, cost–minimizing input demand equations, transformed into cost share equations. By logarithmically differentiating Equation (3) with respect to input prices and employing Shephard's lemma (duality between production and cost functions), the following cost share equations are obtained. According to Shephard's lemma, the input cost share functions can be obtained. Then, we get the following input share equation:

$$S_i = \alpha_i + \sum_{j=1}^{n} \beta_{ij} \ln P_j + \beta_{iy} \ln Y \tag{5}$$

Defining the cost shares $S_i = \frac{P_i X_i}{C}$, it follows that

$$\sum_{i=1}^{n} S_i = 1 \tag{6}$$

where $S_i$ represents the cost share of input $i$. The calculation of price elasticity, which changes with the share of inputs, is shown below. The input-factor demand analysis uses the Allen–Uzawa cross-substitution elasticity $\sigma_{ij}(i \neq j)$, own–price elasticity $\sigma_{ii}$, and price elasticity of demand $\varepsilon_{ij}$ to measure the responses of demand to price changes. The elasticity, as mentioned above, can be estimated by the parameters of input share functions using the following equations:

$$\sigma_{ij} = \frac{(\beta_{ij} + S_i * S_j)}{(S_i * S_j)}(i \neq j) \tag{7}$$

$$\sigma_{ii} = \frac{\beta_{ii} + S_i^2 - S_i}{S_i^2} \tag{8}$$

Similarly, one can derive the own–price elasticities of demand for the *i*th input as

$$\eta_{ii} = \sigma_{ii}S_i \tag{9}$$

Furthermore, cross-price elasticities of factor demand are given by

$$\eta_{ij} = \sigma_{ij}S_j \tag{10}$$

Using Equations (5)–(10), the substitution elasticity is calculated at each point.

*2.2. Data*

This paper investigates the relationship between inputs and outputs of cherry and peach production and constructs simultaneous equations of a translog cost function and share functions. Data were collected from five districts of Beijing (Huairou, Changping, Miyun, Shunyi, and Pinggu) in August 2017. Peach and cherry samples were randomly selected, and 308 valid questionnaires were collected from 39 villages, with 100% validity. Among the samples, there were 150 peach households comprised of 56 picking-garden farmers and 94 traditional farmers, and 158 cherry households comprised of 137 picking–garden farmers and 21 traditional farmers. The inequality number of peach and cherry samples were attributed to the different geographic characteristics of the gardens, but our samples covered around 90% of the total peach and cherry gardens in Beijing, which offers sufficient representation. The survey included the households' and gardens' characteristics, and the input and output factors including costs of labor, fertilizers, manure, pesticides, irrigation, and land. Questionnaires covered two different operation patterns—picking and traditional—both of which involve labor costs, fertilizer costs, and other factors. Different input-factor costs and cost share between picking garden farmers and traditional garden farmers are shown in Table 1.

Table 1 shows that the labor input costs and cost shares for picking gardens were more than those for traditional gardens. However, the fertilizer and pesticide input-factor costs and cost shares for picking gardens were less than that for traditional gardens. Manure and land input-factor costs were also higher for picking gardens. The cost share for labor and land was the highest.

The variables of this study are defined as follows and the descriptions are listed in Table 2:

1.  Labor input and price: This paper chooses labor input per hectare of crop production as the proxy variable for labor input, in terms of the effective working time of workers. Labor price means the labor costs for a day, including workers' wages. The daily wage of household labor was used to calculate the opportunity cost, while workers' wages were used to calculate the cost of employees. This paper uses labor costs per hectare by the number of laborers per hectare to determine the net labor cost.
2.  Fertilizer input and price: Fertilizer input was determined by the amount of fertilizer reduction per hectare, and fertilizer price by dividing fertilizer costs by input.
3.  Manure input and price: To obtain manure price, the cost of manure was divided by the amount of manure.

4.  Pesticide input and price: Pesticide input was calculated as the sum of all pesticide inputs. Pesticide price was obtained by dividing the cost of pesticide by its input amount.
5.  Irrigation input and price: Irrigation price was determined by dividing the cost of irrigation by its input amount.
6.  Average cost: Average cost was calculated as the average of labor cost, fertilizer cost, manure cost, pesticide cost, irrigation cost, and land cost.
7.  Total cost and cost share: Total cost was calculated as the sum of labor cost, fertilizer cost, manure cost, pesticide cost, irrigation cost, and land rentals (capital cost). The total cost share was determined by the ratio of input-factor cost to total cost.

**Table 1.** Input-factor costs and cost share for different operation patterns.

| Variable | Cherry | | Peach | |
|---|---|---|---|---|
| | **Picking Pattern** | **Traditional Pattern** | **Picking Pattern** | **Traditional Pattern** |
| **Input Factors Cost (yuan)** | | | | |
| Labor cost | 7761.51 | 6171.76 | 6901.23 | 5012.19 |
| Fertilizer cost | 36.69 | 41.36 | 39.76 | 46.58 |
| Manure cost | 426.17 | 332.58 | 275.38 | 214.04 |
| Irrigation cost | 101.14 | 122.34 | 130.29 | 227.04 |
| Pesticide cost | 342.46 | 383.63 | 217.08 | 289.94 |
| Land cost | 3319.71 | 3009.52 | 3157.14 | 3017.66 |
| **Cost Share (%)** | | | | |
| Labor share | 0.611 | 0.564 | 0.609 | 0.510 |
| Fertilizer share | 0.003 | 0.004 | 0.004 | 0.006 |
| Manure share | 0.036 | 0.035 | 0.027 | 0.026 |
| Irrigation share | 0.009 | 0.012 | 0.012 | 0.037 |
| Pesticide share | 0.030 | 0.042 | 0.021 | 0.026 |
| Land share | 0.291 | 0.337 | 0.310 | 0.378 |
| Other factor share | 0.021 | 0.006 | 0.018 | 0.017 |

Source: Calculated from own survey data.

**Table 2.** Peach and cherry farmer characteristics and inputs factors.

| Variable | Description | Peach | | Cherry | |
|---|---|---|---|---|---|
| | | **Mean** | **Std. Dev** | **Mean** | **Std. Dev** |
| **Garden Characteristics** | | | | | |
| Operation mode | 0 = common garden; 1 = picking garden | 0.37 | 0.49 | 0.87 | 0.34 |
| **Farmer Characteristics** | | | | | |
| Age (year) | Age of household head | 54.8 | 9.59 | 55.06 | 9.49 |
| Education (year) | Education of household | 8.48 | 3.45 | 8.43 | 2.82 |
| Gender | 0 = female; 1 = male | 0.66 | 0.48 | 0.59 | 0.49 |
| Training | 0 = no; 1 = yes | 0.23 | 0.42 | 0.15 | 0.36 |
| Household size | Number of family members | 4.39 | 1.45 | 4.30 | 1.62 |
| Age of tree (year) | Age of trees | 11.94 | 4.77 | 11.92 | 4.25 |
| **Production Input-Factor Price** | | | | | |
| $P_{labor}$ (yuan/day) | Labor cost divided by the labor inputs | 137.80 | 43.22 | 127.86 | 24.93 |
| $P_{fert}$ (yuan/kg) | Fertilizer cost divided by fertilizer inputs | 2.51 | 0.80 | 2.62 | 0.65 |
| $P_{manure}$ (yuan/kg) | Manure cost divided by manure inputs | 0.18 | 0.39 | 0.16 | 0.13 |
| $P_{water}$ (yuan/m$^3$) | Irrigation cost divided by irrigation inputs | 1.24 | 0.52 | 2.10 | 0.65 |
| $P_{pest}$ (yuan/kg) | Pesticide cost divided by pesticide inputs | 0.05 | 0.03 | 0.07 | 0.08 |
| $P_{land}$ (yuan/ha) | Price of land per ha | 3076 | 538.16 | 3521.65 | 513.15 |
| **Input-Factor Share Function** | | | | | |
| $S_{labor}$ | Labor cost divided by total cost | 0.54 | 0.14 | 0.56 | 0.11 |
| $S_{fert}$ | Fertilizer cost divided by total cost | 0.01 | 0.01 | 0.03 | 0.01 |
| $S_{manure}$ | Manure cost divided by total cost | 0.03 | 0.02 | 0.04 | 0.03 |
| $S_{water}$ | Irrigation cost divided by total cost | 0.02 | 0.14 | 0.01 | 0.01 |
| $S_{pest}$ | Pesticide cost divided by total cost | 0.03 | 0.01 | 0.03 | 0.02 |
| $S_{land}$ | Land cost divided by total cost | 0.37 | 0.12 | 0.33 | 0.09 |

Source: Calculated from own survey data.

This paper used STATA 12.0 software and the survey data of the input-factor prices of cherry and peach production to estimate the cost share functions for each factor. The minimum cost function imposes symmetry and element price uniformity constraints on the relevant parameters. Further, the sum of the cost share of each factor must be equal to 1. Due to the large number of actual production input factors, it is difficult to estimate an overall cost share. Thus, this paper estimates the independent cost share equations for only six factors: labor, fertilizer, manure, irrigation, pesticide, and land.

## 3. Results

To study the impact of different operation patterns on input costs and cost share, the explanatory variables should be continuous variables. From the perspective of simultaneous equations, there is no correlation between the cost function equation and cost share function equations, but the unobservable factors of agricultural production by the same farmers affected both the input cost and cost share, so the error terms were related. Therefore, we use the seemingly unrelated regression (SUR) estimation. SUR can be used when there is no intrinsic relationship between the variables of each equation, but a correlation exists between the error terms of each equation. Thus, we systematically estimate the equations.

According to the different land costs for various farmers, this analysis is divided into two conditions: farmers that face the same land costs, and farmers that do not. If the farmers have the same land cost, this factor has no significant impact on the total production cost. In these cases, the land cost is omitted, and the average production cost and input-factor prices are taken as the explained and explanatory variables, respectively. On the other hand, if farmers have different land costs, this factor has a significant impact on total production cost. In these cases, the total production cost and input-factor prices are taken as the explained and explanatory variables, respectively. Columns (1) and (3) of Table 3 present the parameter estimates of the non-homothetic translog cost function for peach and cherry crops, while columns (2) and (4) present the results without land cost. Detailed results are shown in Appendix A.

Table 3 shows the results of the cost function and input-factor share functions. The result of the cost function shows that the difference in production cost between picking gardens and traditional gardens is significant, and indicates that the production cost of picking gardens is higher than of traditional ones. For the household variable, household size and farmers' age significantly influenced the total input cost. In China, land area is associated with the size of a family, thus a larger household size means a larger distributed land area. Younger farmers generally are well-educated and would like to apply advanced technologies than elder farmers who prefer traditional methods. Technology application increases the total cultivation cost. Regarding the garden characteristics, the age of trees significantly influences the total cost. Cherry trees take 10 years or so for fructification, while peach trees take three to five years. This indicates that if the trees are older, the total production cost is high.

In the labor cost share function, operation pattern has a positive impact on the labor cost share, which indicates that the labor cost of picking gardens is higher than traditional ones. A higher wage rate increases the total labor cost share in cultivation. A higher land price reduces the rented land and labor use. In the fertilizer cost share function, the results show that a higher price of fertilizer reduces the fertilizer use and its cost share. Fertilizer can be substituted with manure, especially in picking gardens; but a higher manure price and labor cost reduce the manure input. A lower manure price gives farmers incentives to substitute fertilizer in their production. In the irrigation cost share function, the coefficient between operation patterns and irrigation cost share is significant and negative, indicating that the irrigation cost of picking gardens is less than of traditional ones. Picking gardens are usually far from mountainous areas and closer to main roads, which have more humid land than traditional gardens A higher water price also reduces irrigation. In the pesticide cost share function, the pesticide cost of picking gardens is more than that of traditional ones. A higher labor cost lowers the pesticide cost share, which is consistent with the results of the labor cost share equation. A higher pesticide price reduces the pesticide use and its cost share. Picking gardens are normally small and scattered, and thus have higher land cost than traditional gardens, which are larger and located near

mountains. Land using manure has higher quality and a higher cost than using fertilizer, but a higher pesticide application will reduce land quality and its price, and the total land-cost share.

**Table 3.** Estimation results of the translog models.

| Variable | (1) Peach (Including Land) | (2) Peach (Excluding Land) | (3) Cherry (Including Land) | (4) Cherry (Excluding Land) |
|---|---|---|---|---|
| Operation mode | 0.21 ** (2.34) | 0.12 ** (2.13) | 0.19 *** (3.20) | 0.11 ** (2.26) |
| **Farmer Characteristics** | | | | |
| Household size | 0.27 *** (2.59) | −0.01 (−0.08) | 0.03 *** (2.49) | −0.03 (−0.34) |
| Age | −0.20 * (−1.87) | −0.16 (−1.30) | −0.18 * (−1.72) | −0.03 (−0.74) |
| **Garden Characteristics** | | | | |
| Age of trees | 0.11 ** (2.35) | 0.04 * (1.77) | 0.05 *** (3.02) | 0.00 ** (2.03) |
| **Input Factors** | | | | |
| $Log(P_{water})$ | 2.90 ** (2.69) | 2.46 *** (4.57) | 0.70 ** (2.93) | 0.22 *** (4.00) |
| $Log(P_{land})$ | 1.92 *** (4.02) | | 2.79 *** (5.30) | |
| $Log(P_{labor})Log(P_{labor})$ | 0.01 (−0.22) | 0.04 ** (2.08) | 0.03 (−1.35) | 0.05 ** (1.97) |
| $Log(P_{labor})Log(P_{fert})$ | −0.02 (−0.35) | 0.05 ** (−2.34) | −0.01 (−0.36) | 0.04 ** (−1.97) |
| $Log(P_{fert})Log(P_{fert})$ | −0.12 (−0.43) | −0.04 * (−1.87) | −0.04 (−0.38) | −0.19 ** (−2.35) |
| $Log(P_{fert})Log(P_{water})$ | −0.19 * (−1.79) | 0.03 (−1.27) | −0.09 * (−1.94) | 0.05 (−1.58) |
| $Log(P_{fert})Log(P_{land})$ | 0.31 * (1.86) | | 0.11 * (1.69) | |
| $Log(P_{manure})Log(P_{manure})$ | −0.04 * (−1.87) | −0.01 (−0.31) | −0.04 * (−1.79) | −0.03 (−1.06) |
| $Log(P_{manure})Log(P_{water})$ | 0.18 * (−1.78) | −0.03 (−1.10) | 0.06 ** (−2.17) | −0.02 (−0.87) |
| $Log(P_{manure})Log(P_{land})$ | −0.26 * (−1.81) | | 0.03 (−0.59) | |
| $Log(P_{water})Log(P_{water})$ | 0.01 *** (−2.63) | 0.00 (−0.41) | 0.00 ** (−2.05) | 0.00 (−0.18) |
| $Log(P_{land})Log(P_{land})$ | −0.07 ** (−2.71) | | −0.17 *** (−4.56) | |
| **Total/Average Yield** | | | | |
| Log(total/average yield) | 1.86 ** (2.05) | 0.66 ** (2.27) | 0.82 *** (3.46) | 0.73 * (1.70) |
| Log(total/average yield) | −0.08 *** (−3.78) | −0.04 (−1.29) | −0.12 *** (−26.75) | 0.03 (−1.06) |
| Log(total/average yield) $Log(P_{water})$ | −0.20 *** (−2.74) | −0.20 *** (−5.78) | −0.02 * (−1.92) | −0.05 ** (−2.27) |
| Log(total/average yield) $Log(P_{land})$ | 0.02 (−0.18) | | 0.10 *** (−3.91) | |
| **$S_{labor}$** | | | | |
| Operation mode | 0.08 *** (3.18) | 0.05 * (1.94) | 0.06 ** (2.54) | 0.05 * (1.92) |
| Log(total/average yield) | 0.05 *** (5.12) | 0.00 (0.07) | 0.03 *** (3.60) | 0.03 (1.47) |
| $Log(P_{labor})$ | 0.04 *** (5.11) | 0.03 *** (3.54) | 0.02 *** (2.86) | 0.01 *** (3.29) |
| $Log(P_{manure})$ | −0.02 (−1.55) | −0.02 (−1.29) | −0.03 ** (−2.45) | −0.02 * (−1.65) |
| $Log(P_{water})$ | −0.04 (−1.09) | −0.01 * (−1.67) | −0.01 (−1.61) | −0.03 * (−1.80) |
| $Log(P_{pest})$ | −0.04 *** (−3.03) | −0.04 *** (−3.29) | −0.00 (−0.50) | −0.00 (−0.12) |
| $Log(P_{land})$ | −0.14 *** (−2.58) | | −0.03 * (−1.74) | |
| Gender | −0.01 (−0.47) | −0.01 * (−1.67) | −0.02 (−0.94) | −0.04 ** (−2.08) |

**Table 3.** *Cont.*

| Variable | (1) Peach (Including Land) | (2) Peach (Excluding Land) | (3) Cherry (Including Land) | (4) Cherry (Excluding Land) |
|---|---|---|---|---|
| $S_{fert}$ | | | | |
| Operation mode | −0.00 (−0.07) | −0.00 (−0.38) | −0.00 (−0.59) | −0.00 (−0.73) |
| Log($P_{fert}$) | 0.02 *** (25.65) | 0.02 *** (25.53) | 0.01 *** (15.97) | 0.01 *** (16.00) |
| Training | −0.00 (−1.12) | −0.00 (−0.76) | −0.00 ** (2.16) | −0.00 ** (2.27) |
| $S_{manure}$ | | | | |
| Operation mode | 0.00 ** (2.13) | 0.00 ** (1.94) | 0.01 ** (1.96) | 0.01 ** (1.99) |
| Log($P_{manure}$) | −0.01 *** (−2.86) | −0.01 *** (−3.02) | −0.00 ** (−2.20) | −0.00 *** (−3.20) |
| Log($P_{pest}$) | 0.01 ** (2.57) | 0.01 *** (2.74) | 0.00 (−0.3) | 0.00 (−0.23) |
| Log(education) | 0.00 (−0.37) | 0.00 (−0.21) | −0.00 *** (−4.20) | −0.00 *** (−4.13) |
| $S_{water}$ | | | | |
| Operation mode | −0.01 *** (−2.74) | −0.01 *** (−2.76) | −0.00 * (−1.92) | −0.00 ** (−2.53) |
| Log(total/average yield) | 0.00 ** (2.28) | 0.00 *** (2.72) | 0.00 ** (2.36) | 0.00 * (1.66) |
| Log($P_{labor}$) | −0.00 ** (−2.07) | −0.00 *** (−3.19) | −0.00 ** (−1.99) | −0.00 ** (−2.07) |
| Log($P_{manure}$) | 0.00 (−1.01) | 0.00 * (−1.81) | 0.00 (−1.41) | 0.00 * (−1.70) |
| Log($P_{water}$) | 0.00 * (1.89) | 0.00 ** (2.20) | 0.00 *** (3.47) | 0.00 *** (3.64) |
| Log($P_{pest}$) | −0.00 *** (−2.91) | −0.00 *** (−2.63) | −0.00 (−1.02) | −0.00 (−0.88) |
| Gender | 0.00 * (1.80) | 0.00 * (1.85) | 0.00 * (−1.71) | 0.00 * (1.90) |
| Log(Age of tree) | 0.01 ** (2.19) | 0.01 ** (2.36) | 0.00 (1.28) | 0.00 (1.45) |
| $S_{pest}$ | | | | |
| Operation mode | −0.01 ** (−2.34) | −0.01 ** (−2.42) | −0.01 ** (−2.49) | −0.01 ** (−2.02) |
| Log(total/average yield) | 0.00 *** (2.76) | 0.00 * (1.72) | 0.00 ** (2.22) | 0.01 ** (1.99) |
| Log($P_{labor}$) | −0.00 *** (−3.85) | −0.00 *** (−3.19) | −0.00 ** (−2.22) | −0.00 ** (−2.51) |
| Log($P_{manure}$) | 0.00 (3.46) | 0.00 (1.81) | 0.00 (1.67) | 0.00 (1.72) |
| Log($P_{pest}$) | −0.00 *** (−3.05) | −0.00 *** (−2.58) | −0.00 * (−1.93) | −0.00 ** (−2.19) |
| Log($P_{land}$) | −0.02 ** (−2.23) | | −0.00 ** (−2.01) | |
| $S_{land}$ | | | | |
| Operation mode | −0.06 *** (−3.13) | | −0.07 *** (−3.38) | |
| Log(total/average yield) | 0.03 *** (4.44) | | 0.02 *** (3.20) | |
| Log($P_{labor}$) | −0.03 *** (−4.95) | | −0.01 ** (−2.52) | |
| Log($P_{fert}$) | −0.00 * (−1.97) | | −0.03 * (−1.94) | |
| Log($P_{manure}$) | 0.02 ** (2.23) | | 0.02 * (1.93) | |
| Log($P_{water}$) | 0.04 *** (3.49) | | 0.00 *** (2.59) | |
| Log($P_{pest}$) | −0.04 *** (−3.45) | | −0.00 ** (−1.95) | |
| Log($P_{land}$) | 0.16 *** (3.36) | | 0.05 *** (2.74) | |
| Sample size | 150 | 150 | 158 | 158 |

The estimation coefficient of the provincial dummy variable is not included. T-values are given in parentheses. ***, **, and * indicate significance at the 1%, 5%, and 10% levels, respectively. Source: Authors' own survey.

As the focus of this paper is to investigate the production input factors for peach and cherry crops in Beijing and determine substitution possibilities, our estimate of elasticities of substitution are limited to substitution possibilities among labor, fertilizer, manure, irrigation, pesticide, and land use in peach and cherry gardens. Based on the results in Table 3, we try to estimate the price elasticity of substitution for these production factors. Our calculations indicate that the price elasticity of all production factors is negative (see Table 4), which implies that the percentage change in their usage quantity is due to a 1% change in their price. From Table 4, we compute the substitution elasticity for various input factors.

**Table 4.** Input price elasticities of peach and cherry.

| Production Factors | Peach | | | Cherry | | |
|---|---|---|---|---|---|---|
| | Total | Picking Pattern | Traditional Pattern | Total | Picking Pattern | Traditional Pattern |
| **Price Elasticity** | | | | | | |
| Labor | −0.570 | −0.576 | −0.590 | −0.506 | −0.505 | −0.513 |
| Fertilizer | −0.869 | −0.820 | −0.837 | −0.808 | −0.830 | −0.889 |
| Manure | −0.235 | −0.218 | −0.246 | −0.420 | −0.414 | −0.427 |
| Irrigation | −0.317 | −0.312 | −0.318 | −0.322 | −0.320 | −0.332 |
| Pesticide | −0.422 | −0.422 | −0.428 | −0.496 | −0.496 | −0.496 |
| **Cross Price Elasticity** | | | | | | |
| Labor and fertilizer | 0.029 | 0.030 | 0.029 | 0.032 | 0.032 | 0.033 |
| Labor and manure | −0.133 | −0.122 | −0.141 | −0.013 | −0.014 | −0.010 |
| Labor and irrigation | 0.295 | 0.275 | 0.308 | 0.004 | 0.003 | 0.005 |
| Labor and Pesticide | −0.070 | −0.069 | −0.071 | 0.037 | 0.036 | 0.042 |
| Fertilizer and manure | 0.440 | 0.494 | 0.502 | 0.248 | 0.256 | 0.207 |
| Fertilizer and irrigation | −0.591 | −0.590 | 0.541 | −0.592 | −0.611 | −0.593 |
| Fertilizer and pesticide | 0.260 | 0.275 | 0.281 | 0.483 | 0.498 | 0.484 |
| Manure and irrigation | 0.260 | 0.285 | 0.309 | 0.872 | 0.862 | 0.838 |
| Manure and pesticide | 0.783 | 0.737 | 0.814 | −0.281 | −0.279 | −0.300 |
| Irrigation and pesticide | 0.296 | 0.372 | 0.264 | −0.023 | −0.025 | −0.029 |

Table 4 shows that the absolute values of price elasticities of input factors are less than 1, and the price elasticities of manure, irrigation, and pesticide are even less than 0.5. Labor cost elasticity is about −0.5; that is, if labor price increases by 10%, labor demand will reduce by 50%. Manure price elasticity is −0.2, meaning that if price increases by 10%, farmers' demand for it decreases by 20%. Irrigation price elasticity is −0.3; that is, if the price increases by 10%, irrigation will reduce by 30%. The price elasticity of pesticides is −0.4; that is, if the price increases by 10%, the pesticide use will reduce by 40%. The absolute value of chemical fertilizer price elasticity is close to 1, and this indicates that an increase of fertilizer price reduces the farmers' demand by 80%. The substitution elasticity is the core index to measure the strength of the substitution relationship between factors. $\sigma_{ij} > 0$ indicates a substitution relationship between factors—the larger the value, the stronger the substitution. By contrast, $\sigma_{ij} < 0$ indicates a complementary relationship between factors. We analyze the substitution elasticity of factors for an in-depth understanding.

First, fertilizer use and labor input had duel relationships. On the one hand, an increased fertilizer use requires more labor input in general, as fertilizer application depends on labor. On the other hand, different fertilizer application practices affect the hours of labor input. A proper fertilizer application method could keep the land nutritious, and thus reduce the labor input on further field management. Therefore, labor input could be more or less for the same amount of fertilizer use. In detail, the economy is the primary consideration behind farmers' decisions, especially in the case of increasing labor costs during production processes. Thus, they might choose to apply fertilizer either many or fewer times, or to increase fertilizer use and reduce manure use, so as to reduce labor costs. Using fertilizer as

a substitute for labor is an ideal choice, but considering environmental pollution, such substitution should be undertaken efficiently.

Second, fertilizer and manure has a substitution relationship, with substitution elasticity more than 0. Substituting manure for fertilizer is a technological innovation, and the farmers' decision to do so is crucial. Substituting manure for fertilizer has high economic benefits, and could also reduce the environmental pollution.

Third, manure and pesticide has a substitution relationship, with substitution elasticity more than 0. Substituting manure for pesticide can alleviate agricultural nonpoint source pollution and reduce farmers' economic costs.

## 4. Conclusions

Leisure agriculture is an industry integrating production, life, and ecology. The purpose of studying leisure agriculture is to combine sightseeing and leisure, promote agricultural transformation, increase agricultural employment, increase farmers' income, and allow the rural economy to prosper. In the context of the rapid development of leisure agriculture in Beijing, picking agriculture is particularly significant. To study the willingness of farmers to choose picking agriculture, we analyzed the input and output of different production factors in cherry and peach gardens. It is noteworthy that input-factor prices were of great importance in farmers' choice of operation. Through a survey of 308 cherry and peach farmers in five districts of Beijing, we analyzed their agriculture-related choices.

Our results showed that there was a large difference between the costs of picking gardens and traditional gardens. Picking gardens had a higher total cost, largely due to their labor-consuming characteristics, with the labor cost share taking a large proportion of the total cost according to our empirical analysis. We found that fertilizer could be substituted with labor. Moreover, manure was a substitution for fertilizer in picking gardens, and farmers used less chemical pesticides under the picking approach. The labor and manure substitution relationship implies that picking gardens could benefit the environment and sustainable agriculture.

In detail, a change in factor price led to a change in the production structure for picking gardens. An increase in factor prices led to a decrease in factor demand, and the factor share decreased. From the empirical results above, the cost shares of labor, irrigation, and pesticide for picking gardens were significant. Furthermore, as labor prices increased, pesticide cost decreased, and an increase in labor price also cause a reduction in land cost share. In peach picking gardens, an increase in pesticide and land prices led to a decrease in labor share. Moreover, as pesticide and labor prices increased, irrigation cost share decreased. This suggests that a change of input cost structure may change the total cost of picking pattern for farmers, and therefore, a way of increasing farmers' income.

The substitution elasticities between labor and fertilizer, fertilizer and manure, and manure and pesticide show that substituting labor with fertilizer and land, and manure with fertilizer and pesticide, has become an important development trend in picking gardens. It implies that the picking pattern is consistent with the development of sustainable agriculture.

The elasticity of different factors also varies. Labor and fertilizer were the most elastic; when labor and fertilizer prices increased by 10%, their demand decreased by 5% and 8%, respectively. There were differences in the elasticity of factors between picking gardens and traditional gardens too, but these differences were not significant. Generally speaking, the elasticity of factors in picking gardens was less than that of factors in traditional gardens. This indicates that production factors in picking gardens were less affected by prices.

We also found some potential issues with picking agriculture in Beijing. Currently, agriculture subsidies from the government mainly target large sightseeing gardens and firms, with few subsidies for small households and picking gardens. Small farmers save costs by investing in labor more than technology or machinery, which impedes the promotion of modern agriculture. Besides, fundamental facilities and advanced farming technologies are not well established and promoted in leisure agriculture.

The government can reduce farmers' production costs by regulating, as needed, the structure of input factors. Minimizing production costs is a prerequisite for increasing farmers' income. More-advanced farming technologies should be extended in suburban areas for leisure agriculture. We also found from our survey that there are more elder farmers operating the picking gardens than younger farmers, but the younger farmers were more well-educated, and easily accepted advanced farming technologies. Thus, the government could encourage younger farmers to devote themselves to leisure agriculture, and offer more agriculture extensions of specific farming skills toward picking agriculture. This could be an additional way of increasing farmers' incomes.

**Author Contributions:** Conceptualization, R.H. and Q.S.; Methodology, analysis and interpretation of results, N.D. and Q.S.; Supervision Q.S. and R.H.; Writing-original draft N.D. and Q.S.; Writing-review& editing Q.S. and N.D. All authors reviewed the results and approved the final version of the manuscript.

**Funding:** This research was supported by Beijing Social Science Foundation (No.15JDJGD020).

**Conflicts of Interest:** The authors declare no conflict of interest.

## Appendix A

**Table A1.** Estimation results of the trans–log models.

| Variable | (1) Peach (Including Land) | (2) Peach (Excluding Land) | (3) Cherry (Including Land) | (4) Cherry (Excluding Land) |
|---|---|---|---|---|
| Constants | 0.65 *** (6.89) | 6.83 *** (2.92) | 0.00 (0.40) | 0.48 ** (2.35) |
| Operation mode | 0.21 ** (2.34) | 0.12 ** (2.13) | 0.19 *** (3.20) | 0.11 ** (2.26) |
| **Farmer Characteristics** | | | | |
| **Training** | 0.02 (−0.27) | −0.08 (−1.30) | 0.06 (−1.1) | −0.01 (−0.22) |
| **Household size** | 0.27 *** (2.59) | −0.01 (−0.08) | 0.03 *** (2.49) | −0.03 (−0.34) |
| **Gender** | 0.03 (−0.49) | −0.02 (−0.49) | −0.02 (−0.56) | −0.03 (−0.74) |
| **Age** | −0.20 * (−1.87) | −0.16 (−1.30) | −0.18 * (−1.72) | −0.03 (−0.74) |
| **Education** | 0.01 (−1.10) | 0.00 (−0.21) | 0.01 (−1.30) | 0.01 (−1.07) |
| **Garden Characteristics** | | | | |
| Age of trees | 0.11 ** (2.35)) | 0.04 * (1.77) | 0.05 *** (3.02) | 0.00 ** (2.03) |
| **Input Factors** | | | | |
| $Log(P_{labor})$ | −0.30 (−0.38) | −0.16 (−0.78) | −0.06 (−0.27) | −0.07 (−0.32) |
| $Log(P_{fert})$ | −2.09 (−1.17) | −0.03 (−0.06) | −0.43 (−0.86) | −0.02 (−0.03) |
| $Log(P_{manure})$ | 2.41 * (−1.87) | −0.07 (−0.27) | −0.43 (−1.02) | 0.49 (−1.39) |
| $Log(P_{water})$ | 2.90 ** (2.69) | 2.46 *** (4.57) | 0.70 ** (2.93) | 0.22 *** (4.00) |
| $Log(P_{pest})$ | 0.17 (−0.18) | −0.20 (−0.77) | −0.14 (−0.55) | 0.24 (−1.02) |
| $Log(P_{land})$ | 1.92 *** (4.02) | | 2.79 *** (5.30) | |
| $Log(P_{labor}) Log(P_{labor})$ | 0.01 (−0.22) | 0.04 ** (2.08) | 0.03 (−1.35) | 0.05 ** (1.97) |

**Table A1.** *Cont.*

| Variable | (1) Peach (Including Land) | (2) Peach (Excluding Land) | (3) Cherry (Including Land) | (4) Cherry (Excluding Land) |
|---|---|---|---|---|
| $Log(P_{labor})$ $Log(P_{fert})$ | −0.02 (−0.35) | 0.05 ** (−2.34) | −0.01 (−0.36) | 0.04 ** (−1.97) |
| $Log(P_{labor})$ $Log(P_{manure})$ | −0.04 (−1.24) | 0.01 (−0.72) | 0.01 (−0.48) | −0.01 (−0.56) |
| $Log(P_{labor})$ $Log(P_{water})$ | 0.07 (−1.41) | 0.01 (−0.41) | −0.00 (−0.05) | 0.01 (−0.57) |
| $Log(P_{labor})$ $Log(P_{pest})$ | −0.03 (−0.60) | 0.00 (−0.03) | 0.00 (−0.07) | 0.00 (−0.13) |
| $Log(P_{labor})$ $Log(P_{land})$ | 0.01 (−0.12) | | −0.03 (−1.12) | |
| $Log(P_{fert})$ $Log(P_{fert})$ | −0.12 (−0.43) | −0.04 * (−1.87) | −0.04 (−0.38) | −0.19 ** (−2.35) |
| $Log(P_{fert})$ $Log(P_{manure})$ | 0.02 (−0.23) | 0.04 (−1.55) | 0.00 (−0.09) | 0.06 (−1.39) |
| $Log(P_{fert})$ $Log(P_{water})$ | −0.19 * (−1.79) | 0.03 (−1.27) | −0.09 * (−1.94) | 0.05 (−1.58) |
| $Log(P_{fert})$ $Log(P_{pest})$ | −0.01 (−0.10) | 0.00 (−0.14) | −0.01 (−0.23) | 0.02 (−0.82) |
| $Log(P_{fert})$ $Log(P_{land})$ | 0.31 * (1.86) | | 0.11 * (1.69) | |
| $Log(P_{manure})$ $Log(P_{manure})$ | −0.04 * (−1.87) | −0.01 (−0.31) | −0.04 * (−1.79) | −0.03 (−1.06) |
| $Log(P_{manure})$ $Log(P_{water})$ | 0.18 * (−1.78) | −0.03 (−1.10) | 0.06 ** (−2.17) | −0.02 (−0.87) |
| $Log(P_{manure})$ $Log(P_{pest})$ | 0.11 (−1.61) | 0.02 (−0.81) | −0.01 (−0.25) | −0.01 (−0.59) |
| $Log(P_{manure})$ $Log(P_{land})$ | −0.26 * (−1.81) | | 0.03 (−0.59) | |
| $Log(P_{water})$ $Log(P_{water})$ | 0.01 *** (−2.63) | 0.00 (−0.41) | 0.00 ** (−2.05) | 0.00 (−0.18) |
| $Log(P_{water})$ $Log(P_{pest})$ | −0.02 (−0.23) | −0.01 (−0.24) | −0.00 (−0.08) | −0.00 (−0.21) |
| $Log(P_{water})$ $Log(P_{land})$ | −0.09 (−0.44) | | −0.07 (−1.59) | |
| $Log(P_{pest})$ $Log(P_{pest})$ | −0.03 (−1.01) | −0.01 (−1.23) | 0.00 (−0.47) | −0.01 (−0.73) |
| $Log(P_{pest})$ $Log(P_{land})$ | −0.06 (−0.54) | | −0.01 (−0.23) | |
| $Log(P_{land})$ $Log(P_{land})$ | −0.07 ** (−2.71) | | −0.17 *** (−4.56) | |
| **Total/average Yield** | | | | |
| $Log(total/average\ yield)$ | 1.86 ** (2.05) | 0.66 ** (2.27) | 0.82 *** (3.46) | 0.73 * (1.70) |
| $Log(total/average\ yield)$ | −0.08 *** (−3.78) | −0.04 (−1.29) | −0.12 *** (−26.75) | 0.03 (−1.06) |
| $Log(total/average\ yield)$ $Log(P_{labor})$ | 0.03 (−1.27) | 0.00 (−0.07) | 0.02 ** (−2.40) | 0.05 * (−1.76) |
| $Log(total/average\ yield)$ $Log(P_{fert})$ | −0.02 (−0.20) | 0.04 (−0.72) | −0.04 (−1.58) | 0.04 (−0.58) |
| $Log(total/average\ yield)$ $Log(P_{manure})$ | 0.00 (−0.05) | −0.01 (−0.21) | 0.00 (−0.28) | −0.07 (−1.37) |
| $Log(total/average\ yield)$ $Log(P_{water})$ | −0.20 *** (−2.74) | −0.20 *** (−5.78) | −0.02 * (−1.92) | −0.05 ** (−2.27) |
| $Log(total/average\ yield)$ $Log(P_{pest})$ | 0.05 (−0.54) | 0.01 (−0.26) | 0.02 (−1.54) | 0.05 (−1.38) |
| $Log(total/average\ yield)$ $Log(P_{land})$ | 0.02 (−0.18) | | 0.10 *** (−3.91) | |

**Table A1.** *Cont.*

| Variable | (1) Peach (Including Land) | (2) Peach (Excluding Land) | (3) Cherry (Including Land) | (4) Cherry (Excluding Land) |
|---|---|---|---|---|
| | | $S_{labor}$ | | |
| Operation mode | 0.08 *** | 0.05 * | 0.06 ** | 0.05 * |
| | (3.18) | (1.94) | (2.54) | (1.92) |
| Log(total/average yield) | 0.05 *** | 0.00 | 0.03 *** | 0.03 |
| | (−5.12) | (−0.07) | (−3.60) | (−1.47) |
| Log($P_{labor}$) | 0.04 *** | 0.03 *** | 0.02 *** | 0.01 *** |
| | (5.11) | (3.54) | (2.86) | (3.29) |
| Log($P_{fert}$) | −0.01 | −0.01 | 0.03 * | 0.03 |
| | (−0.38) | (−0.31) | (−1.66) | (−1.52) |
| Log($P_{manure}$) | −0.02 | −0.02 | −0.03 ** | −0.02 * |
| | (−1.55) | (−1.29) | (−2.45) | (−1.65) |
| Log($P_{water}$) | −0.04 | −0.01 * | −0.01 | −0.03 * |
| | (−1.09) | (−1.67) | (−1.61) | (−1.80) |
| Log($P_{pest}$) | −0.04 *** | −0.04 *** | −0.00 | −0.00 |
| | (−3.03) | (−3.29) | (−0.50) | (−0.12) |
| Log($P_{land}$) | −0.14 *** | | −0.03 * | |
| | (−2.58) | | (−1.74) | |
| **Training** | −0.02 | −0.00 | 0.02 | 0.02 |
| | (−0.65) | (−0.03) | (−0.73) | (−0.93) |
| Log(Household size) | 0.03 | 0.00 | −0.01 | −0.01 |
| | (−0.94) | (−0.03) | (−0.26) | (−0.55) |
| **Log(Age)** | −0.04 | −0.06 | −0.04 | −0.03 |
| | (−0.71) | (−1.03) | (−0.90) | (−0.79) |
| **Log(Education)** | 0.00 | 0.00 | 0.00 | 0.00 |
| | (−0.98) | (−0.25) | (−1.23) | (−1.43) |
| Gender | −0.01 | −0.01 * | −0.02 | −0.04 ** |
| | (−0.47) | (−1.67) | (−0.94) | (−2.08) |
| **Log(Age of tree)** | −0.01 | −0.02 | 0.01 | 0.00 |
| | (−0.32) | (−0.91) | (0.60) | (0.13) |
| | | $S_{fert}$ | | |
| Operation mode | −0.00 | −0.00 | −0.00 | −0.00 |
| | (−0.07) | (−0.38) | (−0.59) | (−0.73) |
| Log(total/average yield) | 0.0 | 0.00 | −0.00 | 0.00 |
| | (−0.45) | (−0.22) | (−0.07) | (−0.12) |
| Log($P_{labor}$) | −0.04 | −0.06 | −0.02 | −0.04 |
| | (−0.90) | (−0.88) | (−0.76) | (−0.82) |
| Log($P_{fert}$) | 0.02 *** | 0.02 *** | 0.01 *** | 0.01 *** |
| | (25.65) | (25.53) | (15.97) | (16.00) |
| Log($P_{manure}$) | 0.00 | 0.00 | −0.00 | −0.00 |
| | (−1.54) | (−1.24) | (−1.22) | (−1.19) |
| Log($P_{water}$) | 0.00 | 0.00 | −0.00 | −0.00 |
| | (−0.08) | (−0.2) | (−0.75) | (−0.77) |
| Log($P_{pest}$) | 0.00 | 0.00 | 0.00 | 0.00 |
| | (−0.07) | (−0.14) | (−0.8) | (−0.8) |
| Log($P_{land}$) | −0.00 | | −0.00 | |
| | (−0.79) | | (−0.8) | |
| **Training** | −0.00 | −0.00 | −0.00 ** | −0.00 ** |
| | (−1.12) | (−0.76) | (2.16) | (2.27) |
| Log(Household size) | −0.00 | −0.00 | 0.00 | 0.00 |
| | (−0.26) | (−0.29) | (−1.05) | (−1) |
| **Log(Age)** | 0.00 | 0.00 | −0.00 | −0.00 |
| | (−0.71) | (−0.65) | (−0.42) | (−0.44) |
| **Log(Education)** | 0.00 | 0.00 | −0.00 | −0.00 |
| | (−0.31) | (0.41) | (−0.64) | (−0.61) |
| **Gender** | 0.00 | 0.00 | −0.00 | −0.00 |
| | (−1.28) | (−1.27) | (−0.11) | (−0.15) |
| Log(Age of tree) | −0.01 | 0.00 | −0.00 | 0.00 |
| | (−0.32) | (0.63) | (−0.65) | (−0.69) |

**Table A1.** *Cont.*

| Variable | (1) Peach (Including Land) | (2) Peach (Excluding Land) | (3) Cherry (Including Land) | (4) Cherry (Excluding Land) |
|---|---|---|---|---|
| | $S_{manure}$ | | | |
| Operation mode | 0.00 ** (2.13) | 0.00 ** (1.94) | 0.01 ** (1.96) | 0.01 ** (1.99) |
| Log(total/average yield) | 0.00 (−1.46) | 0.00 (−0.2) | 0.00 (−0.54) | 0.00 (−0.27) |
| Log($P_{labor}$) | −0.00 (−1.61) | −0.00 (−1.28) | −0.00 (−0.14) | 0.00 (−0.22) |
| Log($P_{fert}$) | −0.00 (−0.83) | −0.00 (−0.77) | −0.01 (−0.69) | −0.01 (−1.55) |
| Log($P_{manure}$) | −0.01 *** (−2.86) | −0.01 *** (−3.02) | −0.00 ** (−2.20) | −0.00 *** (−3.20) |
| Log($P_{water}$) | −0.00 (−0.59) | −0.00 (−0.19) | 0.00 (−0.47) | 0.00 (−0.46) |
| Log($P_{pest}$) | 0.01 ** (2.57) | 0.01 *** (2.74) | 0.00 (−0.3) | 0.00 (−0.23) |
| Log($P_{land}$) | −0.00 (−0.20) | | −0.02 (−1.01) | |
| **Training** | −0.00 (−0.39) | −0.00 (−0.36) | −0.01 (−0.94) | −0.00 (−0.79) |
| Log(Household size) | 0.00 (−0.45) | −0.00 (−0.11) | 0.00 (−0.77) | 0.00 (−0.63) |
| **Log(Age)** | 0.01 (−0.68) | −0.00 (−0.38) | 0.02 (−1.46) | 0.01 (−1.35) |
| **Log(Education)** | 0.00 (−0.37) | 0.00 (−0.21) | −0.00 *** (−4.20) | −0.00 *** (−4.13) |
| **Gender** | 0.00 (−1.17) | 0.00 (−1.62) | 0.00 (−0.57) | 0.00 (−0.73) |
| **Log(Age of tree)** | 0.00 (0.01) | 0.00 (0.17) | −0.01 (−1.44) | −0.01 (−1.47) |
| | $S_{water}$ | | | |
| Operation mode | −0.01 *** (−2.74) | −0.01 *** (−2.76) | −0.00 * (−1.92) | −0.00 ** (−2.53) |
| Log(total/average yield) | 0.00 ** (2.28) | 0.00 *** (2.72) | 0.00 ** (2.36) | 0.00 * (1.66) |
| Log($P_{labor}$) | −0.00 ** (−2.07) | −0.00 *** (−3.19) | −0.00 ** (−1.99) | −0.00 ** (−2.07) |
| Log($P_{fert}$) | −0.00 (−1.42) | 0.00 (−0.35) | −0.00 (−0.35) | −0.00 (−0.21) |
| Log($P_{manure}$) | 0.00 (−1.01) | 0.00 * (−1.81) | 0.00 (−1.41) | 0.00 * (−1.70) |
| Log($P_{water}$) | 0.00 * (1.89) | 0.00 ** (2.20) | 0.00 *** (3.47) | 0.00 *** (3.64) |
| Log($P_{pest}$) | −0.00 *** (−2.91) | −0.00 *** (−2.63) | −0.00 (−1.02) | −0.00 (−0.88) |
| Log($P_{land}$) | −0.00 (−0.16) | | 0.00 (−0.2) | |
| **Training** | 0.00 (−0.49) | 0.00 (−0.53) | −0.00 (−0.35) | −0.00 (−0.60) |
| Log(Household size) | −0.00 (−0.42) | −0.00 (−0.11) | −0.00 (−1.12) | −0.00 (−0.90) |
| **Log(Age)** | −0.00 (−0.51) | −0.00 (−0.38) | −0.00 (−0.33) | −0.00 (−0.66) |
| **Log(Education)** | −0.00 (−0.51) | −0.00 (−0.21) | 0.00 (−0.35) | 0.00 (−0.46) |
| **Gender** | 0.00 * (1.80) | 0.00 * (1.85) | 0.00 * (−1.71) | 0.00 * (1.90) |
| **Log(Age of tree)** | 0.01 ** (2.19) | 0.01 ** (2.36) | 0.00 (1.28) | 0.00 (1.45) |

**Table A1.** *Cont.*

| Variable | (1) Peach (Including Land) | (2) Peach (Excluding Land) | (3) Cherry (Including Land) | (4) Cherry (Excluding Land) |
|---|---|---|---|---|
| | **$S_{pest}$** | | | |
| Operation mode | −0.01 ** (−2.34) | −0.01 ** (−2.42) | −0.01 ** (−2.49) | −0.01 ** (−2.02) |
| Log(total/average yield) | 0.00 *** (2.76) | 0.00 * (1.72) | 0.00 ** (2.22) | 0.01 ** (1.99) |
| Log($P_{labor}$) | −0.00 *** (−3.85) | −0.00 *** (−3.19) | −0.00 ** (−2.22) | −0.00 ** (−2.51) |
| Log($P_{fert}$) | 0.00 (−0.03) | 0.00 (−0.35) | −0.00 (−0.87) | −0.00 (−0.83) |
| Log($P_{manure}$) | 0.00 ** (3.46) | 0.00 * (1.81) | 0.00 * (1.67) | 0.00 * (1.72) |
| Log($P_{water}$) | 0.00 (−0.95) | 0.00 (−0.07) | 0.00 (−0.62) | 0.00 (−0.75) |
| Log($P_{pest}$) | −0.00 *** (−3.05) | −0.00 *** (−2.58) | −0.00 * (−1.93) | −0.00 ** (−2.19) |
| Log($P_{land}$) | −0.02 ** (−2.23) | | −0.00 ** (−2.01) | |
| **Training** | 0.00 (−0.62) | 0.00 (−1) | 0.00 (−0.86) | 0.00 (−0.84) |
| Log(Household size) | −0.00 (−1.48) | 0.00 (−1.17) | 0.00 (−0.46) | 0.00 (−0.6) |
| **Log(Age)** | 0.01 (−1.34) | 0.01 (−1.29) | 0.01 (−0.79) | 0.01 (−0.83) |
| **Log(Education)** | −0.00 (−1.61) | −0.00 (−1.04) | −0.00 (−0.24) | −0.00 (−0.52) |
| **Gender** | 0.00 (−0.67) | 0.00 (−0.69) | 0.00 (−1.1) | 0.01 (0.92) |
| **Log(Age of tree)** | −0.00 (−0.09) | 0.00 (0.15) | 0.00 (0.40) | 0.00 (0.71) |
| | **$S_{land}$** | | | |
| Operation mode | −0.06 *** (−3.13) | | −0.07 *** (−3.38) | |
| Log(total/average yield) | 0.03 *** (4.44) | | 0.02 *** (3.20) | |
| Log($P_{labor}$) | −0.03 *** (−4.95) | | −0.01 ** (−2.52) | |
| Log($P_{fert}$) | −0.00 * (−1.97) | | −0.03 * (−1.94) | |
| Log($P_{manure}$) | 0.02 ** (2.23) | | 0.02 * (1.93) | |
| Log($P_{water}$) | 0.04 *** (3.49) | | 0.00 *** (2.59) | |
| Log($P_{pest}$) | 0.04 *** (3.45) | | 0.00 ** (1.95) | |
| Log($P_{land}$) | 0.16 *** (3.36) | | 0.05 *** (2.74) | |
| **Training** | −0.01 (−0.58) | | −0.02 (−1.19) | |
| Log(Household size) | −0.01 (−0.58) | | 0.01 (−0.52) | |
| **Log(Age)** | 0.04 (−0.94) | | 0.03 (−0.79) | |
| **Log(Education)** | −0.00 (−0.93) | | −0.00 (−0.29) | |
| **Gender** | 0.00 (−0.2) | | 0.00 (−0.35) | |
| **Log(Age of tree)** | 0.01 (0.40) | | −0.01 (−0.65) | |
| **Sample size** | 150 | 150 | 158 | 158 |

***, **, and * indicate significance at the 1%, 5%, and 10% levels, respectively.

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
