# Peer review of "Price Elasticity of Production Factors in Beijing’s Picking Gardens"

_sustainability, doi:10.3390/su11072160_

Round 1
Reviewer 1 Report
From the scientific point of view the paper appears valid.
The theme is very timely for the development of Price elasticity of production factors in Beijing’s picking gardens. The topic is of great interest and relevance.
Some suggestions seen to be necessary to improve the paper.
Especially Introduction part, the paper should better provide a critical analysis of the available and appropriate literature.
The bibliography is extensive and relevant with an invitation to see her again, considering those jobs that literature reports on Mediterranean products and by-products that could widely used in biogas plants
From the scientific point of view the work appears valid.
Author Response
Dear Sir or Madam,
We are grateful for the thorough and helpful review of our journal submission you kindly provided. We feel that the comments have greatly improved our thinking about the issues. Please find below our best effort to respond to your concerns item by item in the italicized script.
Referee Report for the Manuscript
Price elasticity of production factors in Beijing’s picking gardens
submitted to
Sustainability-469008
Comments and Suggestions for Authors:
From the scientific point of view the paper appears valid.
The theme is very timely for the development of Price elasticity of production factors in Beijing’s picking gardens. The topic is of great interest and relevance.
Some suggestions seen to be necessary to improve the paper.
[1] Especially Introduction part, the paper should better provide a critical analysis of the available and appropriate literature.
Thank you very much for the comments. We rewrote the introduction part and added additional literature for leisure agriculture and methodology. Please find it in line 47-86 in the “no markup” version of the paper.
[2] The bibliography is extensive and relevant with an invitation to see her again, considering those jobs that literature reports on Mediterranean products and by-products that could widely used in biogas plants
Thanks a lot for the comments. We added additional related literature to the text as we replied in question [1], and we also considered your advice and tried to cooperate with the current and further studies.
From the scientific point of view the work appears valid.

Reviewer 2 Report
The introduction and, above all, the conclusions can be improved in order to show better aim and results for further studies in the topic.
Author Response
Dear Sir or Madam,
We are grateful for the thorough and helpful review of our journal submission you kindly provided. We feel that the comments have greatly improved our thinking about the issues. Please find below our best effort to respond to your concerns item by item in the italicized script.
Referee Report for the Manuscript
Price elasticity of production factors in Beijing’s picking gardens
submitted to
Sustainability-46900
Comments and Suggestions for Authors:
The introduction and, above all, the conclusions can be improved in order to show better aim and results for further studies in the topic.
Thanks a lot for your comments. We rewrote the introduction and conclusion part. We included more related literature and offered more in-depth discussion and issues for further studies in the conclusion. Please find it in the revised version.

Reviewer 3 Report
• The paper deal with an interesting topic of price elasticity in agriculture. Furthermore, it concerns the leisure agriculture which is relatively rarely addressed problem in literature.
• The introduction is generally well written, however the literature review concerning research on elasticities is rather superficial so it can be extended. At least, the authors should write something more about the research they quote. For example, in lines 64-69 they provide some conclusion but we don’t know where, when and on which sample the study was conducted.
• The novelty of the adopted approach is given, however the aim seems to be not very clear. Are the authors really study the impact of production factors on agricultural economic growth? It would be better to write exactly what was done.
• Second and third part of the paper could be merged in ‘data and methods’. Then the authors may divide it into two components: functional form used and data description.
• for the variables that take the values 0 or 1 (table 2) it would be better to put the numbers ( e.g how many farmers follow traditional pattern) than mean and SD. At minimum, both information should be provided
• the fourth part should be rather ‘results’, not ‘discussion’, especially because the authors present only their own estimation.
• The first paragraph in part 4 could be moved to the previous section
• In my opinion, the results in the fourth part may be interesting but in their present form they are not well communicated. My opinion is based on several arguments: firstly, the explanation of SUR method is missing and the readers who are not familiar with this method may have problem with understanding results in table 3; secondly, there are many superficial and imprecise phrases (especially in lines 210-223, e.g production costs of picking gardens for peach and cherry crops are 210 significant, line 216-18 etc.)which may result also with vaguely English used sometimes as well as misspelling; thirdly, there is very little information about farmer and garden characteristics and their impact on total cost; fourthly, are 150 observation enough for estimation that employ so many variables as demonstrated in annex?; fifthly, the authors should indicate more clearly how the table number 3 should be read.
• It seems that also in substitution elasticity result part there are some mistakes. E.g in line 240-241 the authors write The results also showed that cherry and peach producers would not reduce the input of labor and fertilizer because of an increase in their prices; the elasticity of labor and fertilizer were higher. Is this right? These elasticities are closer to one so it means that increase in their prices lead to relatively large decrease in demand, doesn't it?
• In lines 248-250 they write that labor and fertilisers have a substitution relationship but the inpretation suggest that this is complementary relations. Similar problem may be found in lines 261-263.
• Taking above reservations, the result part should be rewritten and language should be checked.
• In conclusions there is mistake in lines 282-283 (two times irrigation share).
• The policy recommendation are very scarce. Could you extend it a bit?
Author Response
Dear Sir or Madam,
We are grateful for the thorough and helpful review of our journal submission you kindly provided. We feel that the comments have greatly improved our thinking about the issues. Please find below our best effort to respond to your concerns item by item in the italicized script.
Referee Report for the Manuscript
Price elasticity of production factors in Beijing’s picking gardens
submitted to
Sustainability-469008
Comments and Suggestions for Authors:
• The paper deal with an interesting topic of price elasticity in agriculture. Furthermore, it concerns the leisure agriculture which is relatively rarely addressed problem in literature.
[1] The introduction is generally well written, however the literature review concerning research on elasticities is rather superficial so it can be extended. At least, the authors should write something more about the research they quote. For example, in lines 64-69 they provide some conclusion but we don’t know where, when and on which sample the study was conducted.
Thank you very much for the comments. We rewrote the introduction part and added more related literature to the topic and methodology (i.e. elasticities). We also sorry for the unclear literature in the previous version, and we traced every literature we cited and put them in the references.
[2] The novelty of the adopted approach is given, however the aim seems to be not very clear. Are the authors really study the impact of production factors on agricultural economic growth? It would be better to write exactly what was done.
Thanks a lot for your comments. We apologize for our ambiguous words in the previous version. We changed the wording and interpretation, and make our research aim clear now. We offered our research aim in line 87-97 in the “no markup” version.
[3] Second and third part of the paper could be merged in ‘data and methods’. Then the authors may divide it into two components: functional form used and data description.
Thank you for your suggestion. We changed the organization.
[4] for the variables that take the values 0 or 1 (table 2) it would be better to put the numbers ( e.g how many farmers follow traditional pattern) than mean and SD. At minimum, both information should be provided
Thanks for the suggestion. We explained this in line 156-161 in the ‘no markup’ version.
[5] the fourth part should be rather ‘results’, not ‘discussion’, especially because the authors present only their own estimation.
Thank you for your suggestion. We changed the section title.
[6] The first paragraph in part 4 could be moved to the previous section
Thank you. We did as you suggested.
[7] In my opinion, the results in the fourth part may be interesting but in their present form they are not well communicated. My opinion is based on several arguments: firstly, the explanation of SUR method is missing and the readers who are not familiar with this method may have problem with understanding results in table 3; secondly, there are many superficial and imprecise phrases (especially in lines 210-223, e.g production costs of picking gardens for peach and cherry crops are 210 significant, line 216-18 etc.)which may result also with vaguely English used sometimes as well as misspelling; thirdly, there is very little information about farmer and garden characteristics and their impact on total cost; fourthly, are 150 observation enough for estimation that employ so many variables as demonstrated in annex?; fifthly, the authors should indicate more clearly how the table number 3 should be read.
Thanks a lot for the detailed comments. (1) we explained the SUR method and the reason that we use this method in line 205-211 in ‘no markup’ version. (2) we are sorry for our poor writing in the last version. We rewrote the result part and the result explanation. (3) We explained more about this in line 228-234 in ‘no markup’ version. (4) we totally have 308 samples, 150 or so each. We know that it is not a large sample size for microeconomics analysis, but it covers 90% of the total peach and cherry farmers for leisure agriculture in Beijing. We explained this in line 161-167 in ‘no markup’ version. (5) Yes. We explained more about table 3 in line 225-252.
[8] It seems that also in substitution elasticity result part there are some mistakes. E.g in line 240-241 the authors write The results also showed that cherry and peach producers would not reduce the input of labor and fertilizer because of an increase in their prices; the elasticity of labor and fertilizer were higher. Is this right? These elasticities are closer to one so it means that increase in their prices lead to relatively large decrease in demand, doesn't it?
Thank you very much for your detailed comments. We rewrote the elasticity part and corrected those mistakes.
[9] In lines 248-250 they write that labor and fertilisers have a substitution relationship but the inpretation suggest that this is complementary relations. Similar problem may be found in lines 261-263.
Thank you very much. We made typos there and we changed it in the updated version.
[10] Taking above reservations, the result part should be rewritten and language should be checked.
Thanks for the comments. We rewrote the result part with your comments and we checked our language.
[11] In conclusions there is mistake in lines 282-283 (two times irrigation share).
Thanks. We changed it.
[12] The policy recommendation are very scarce. Could you extend it a bit?
Yes. We add additional policy recommendation in the last paragraph of the conclusion, in line 332-339 in ‘no markup’ version.

Round 2
Reviewer 3 Report
The authors has addressed almost all my remarks. Still there are somr minor shortcomings:
- In Table 1 please add the units
- in lines 272-275 there is still inconsistency: you write ‘The more the use of fertilizer, the more the labor input needed. Thus, both were 'substitution.’ So you call this as the 'substitution' but your description is as 'complimentary effect'. Please check carefully the interpretation of this part of the result.
Author Response
Referee Report for the Manuscript
Price elasticity of production factors in Beijing’s picking gardens
submitted to
Sustainability-469008
Dear Sir or Madam,
We are grateful for the additional comments. Please find below our best effort to respond to your concerns item by item in the italicized script.
Comments and Suggestions for Authors:
The authors has addressed almost all my remarks. Still there are somr minor shortcomings:
- In Table 1 please add the units
Thank you for your suggestion. We have added the units.
- in lines 272-275 there is still inconsistency: you write ‘The more the use of fertilizer, the more the labor input needed. Thus, both were 'substitution.’ So you call this as the 'substitution' but your description is as 'complimentary effect'. Please check carefully the interpretation of this part of the result.
Thanks a lot for your detailed comments. We apologize that we made you feel confused. We intended to explain the relations between fertilizer and labor input like this. On one hand, more fertilizer use requires more labor input in general for fertilizer application has to use labor. On the other hand, different fertilizer application practices affect the hours of labor input. A proper fertilizer application method could keep the land nutritious, and thus reduces the labor input on further field management. Therefore, labor input could be less or more for the same amount of fertilizer use. We have changed the corresponding part of the paper in line 271-275.
